# Erasure Codes for Cold Data in Distributed Storage Systems

Chao Yin [1,2], Zhiyuan Xu [2,1], Wei Li [1], Tongfang Li [1,*], Sihao Yuan [1] and Yan Liu [1]

1 School of Computer and Big Data Science, Jiujiang University, Jiujiang 332005, China
2 School of Information Management, Jiangxi University of Finance and Economics, Nanchang 330013, China
* Correspondence: maggie0611@163.com

**Abstract:** Replication and erasure codes are always used for storing large amounts of data in distributed storage systems. Erasure code technology can maximize the storage space of distributed storage systems as well as guaranteeing their availability and reliability, but it will decrease the performance of the system when encoding and decoding. Since cold data do not require high real-time data availability, we focus on the cold data using erasure codes. We propose a new erasure code process named NewLib code based on the Liberation code, which designs the data alignment after stripping the encoding data. The NewLib code improves the performance of reading and writing in the distributed storage systems. At the same time, we developed a node scheduling scheme called N-Schedule, which divides the data nodes into multiple virtual nodes according to the storage space and computing power. The virtual nodes are dispersed into a hash ring by a consistent hash to construct a fully symmetric and decentralized hash ring in order to achieve uniform data distribution and task scheduling. The experimental results show our scheme can improve the performance of the system.

**Keywords:** erasure codes; NewLib code; N-Schedule scheme; performance

## 1. Introduction

Cloud storage combines a number of storage devices on the network using a distributed file system. Many companies use cloud storage services to jointly provide data storage and service access functions [1–3]. The reliability of a distributed storage system includes two aspects [4]: the availability and the durability of the data. Cloud storage systems improve the data availability and the storage system reliability through optimization management, data recovery, and so on. Data reliability is primarily dependent on data redundancy [5,6]. Fault tolerance is used to preserve the availability of the data in the event of system damage or other specific circumstances.

Currently, two commonly used data redundancy strategies are replication technology [7–9] and erasure code technology. Replication technology involves keeping multiple replicas of a data item on multiple nodes distributed throughout a system to improve system reliability and access efficiency. Unlike replication technology, with erasure code technology, the original data are first divided into m parts, then transformed into n (n > m) parts. When recovering the data, the complete data can be restored as long as any part of t (t ≥ m) is obtained.

The Google cloud storage service system uses replication technology to ensure data reliability. The data blocks have three replicas by default in the GFS, and users can also make different replicas for different file systems. The GFS ensures maximum data reliability and availability and maximum network bandwidth utilization, through a series of replica placement and replica tuning policies. However, the storage efficiency of the three replicas is relatively inefficient; so, many cloud storage companies are considering using an erasure code strategy to improve it.

If we classify all the data in a big data system, we find that the importance of these data is not equivalent. Some data are used often with high real-time requirements, while other data may only be used once after a long time, and some may not be used at all after

storage. These data are called cold data. We propose a storage schedule scheme combining the advantage of the erasure code strategy and the replication strategy to reduce the data redundancy.

The contributions of this paper are described as follows:

- We propose a storage schedule scheme combining the advantage of the erasure code strategy and the replica strategy to save the storage space that cold data use.
- We propose the NewLib code scheme and the N-Schedule strategy to improve the encoding and decoding speed. At the same time, we successfully designed and implemented a node schedule to increase the data addressing performance.

The rest of this paper is organized as follows: The related works are described in Section 2. The theory of our scheme and its implementation are presented in Sections 3 and 4. Section 5 gives the experimental results and evaluation. The conclusion follows in Section 6.

## 2. Related Works

### 2.1. Research on Cloud Storage System Architecture

Cloud storage services adopt a master/slave structure, which is the central storage architecture. For example, the infrastructure of Google is mainly composed of four parts: the Google File System [10], the parallel data processing model Map/Reduce [11], the structured data table BigTable [12], and the distributed lock Chubby, as shown in Figure 1a.

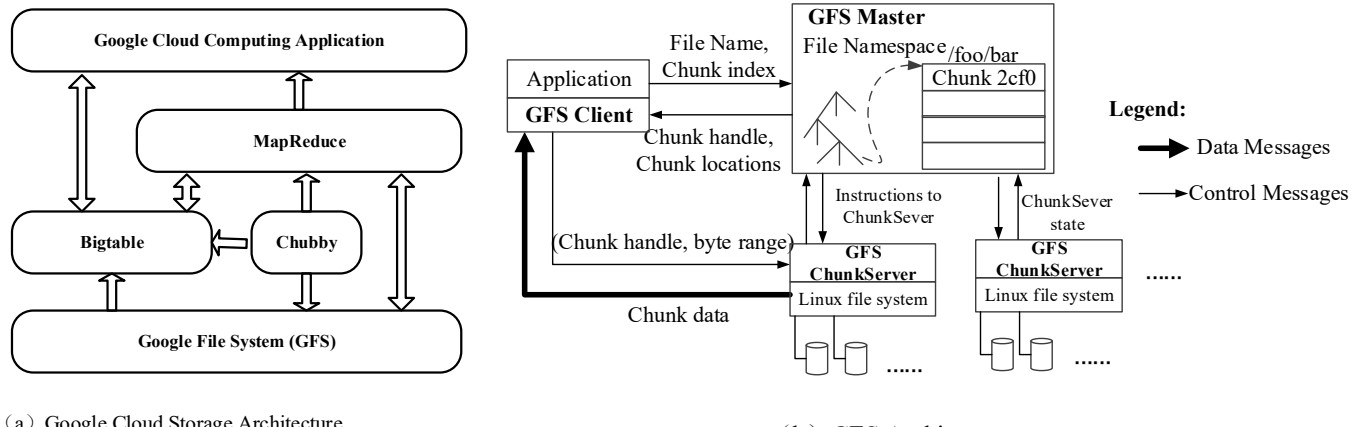

（a）Google Cloud Storage Architecture

（b）GFS Architect

**Figure 1.** Google cloud storage system architecture.

In Figure 1b, each GFS cluster is constituted by multiple chunk servers and one master. The chunk servers store the block data, and the master stores the metadata of the file system. When the system needs to access the file, it communicates with the master to obtain the chunk server information of the file and the location information of the file replica. Then, it directly communicates with the chunk server to obtain the corresponding file. Yahoo's HDFS [13], Amazon's S3 [14], Microsoft's SkyDrive, IBM's Blue Cloud, and KFS [15] also use this architecture.

This single storage system has high performance with central storage architecture, which is suitable for providing dedicated storage services. The disadvantage is that centralization brings a single point of failure, which makes the system unable to cope with service interruptions caused by unforeseen sudden failures, and many data centers are unable to cope with failures [16]. For example, in February 2011, Gmail was unavailable to 0.02% of users due to a failure.

### 2.2. Decentralized Architecture

The typical decentralized structure is peer-to-peer (P2P) storage technology, which refers to the technology of establishing storage networks in a functionally equivalent manner between storage nodes. In peer-to-peer technology, many server nodes compose

the peer-to-peer storage system, which can be divided into a closed system and an open system, according to the environment to which they adapt [17].

The nodes are relatively stable in the closed system, which has strict central authentication, auditing, and management functions to ensure the continuous operation of the nodes. Although there may also be temporary errors in the closed system, it will be repaired in time. The nodes will not quit the system at any time, and they will cooperate very well.

Amazon's Dynamo [18] is a classic example. Amazon has established dozens of large data centers around the world, routing requests from the local to the corresponding nodes through the Gossip protocol. They use a modified consistency hash algorithm to determine the range of nodes where the data are stored and assign this hash to each node on the ring. A value in the range represents its position on the ring, and then the data key is hashed to determine the node it stores. The decentralized storage system has good scalability and will not cause the whole system to collapse due to the failure of a single server. In addition, Microsoft uses a location service [19] to solve the problem of routing in multiple data centers.

In an open system, nodes can join or quit at any time without guaranteeing continuous online service delivery, which leads to temporary errors and permanent errors more frequently than closed systems. The degree of collaboration between nodes is low, and there can be many rational users. A typical representative of such a system is a P2P storage system composed of personal computers.

*2.3. The Array Codes*

Array codes use an exclusive OR operation for code verification, and they have a higher encoding speed and decoding speed. Not all array codes are MDS codes. Only the array codes whose code length and minimum distance reach the singleton boundary belong to MDS codes. The MDS array codes can achieve the highest storage efficiency, which means the storage redundancy is the lowest. The array erasure code can be divided into horizontal coding and vertical coding according to the distribution form of the data chunk and the parity chunk.

Compared with RS codes, array codes are completely based on an XOR operation, which is a hot topic in erasure code research. The purpose of the array codes is to store the original data and redundancy in a two-dimensional (or multidimensional) array [20]. Compared with the traditional RS codes, only the XOR operation is used in the array code, which is easy to implement. At the same time, the processes of coding, updating, and reconstruction are relatively simple, making the application the most widely used. The array codes are divided into horizontal parity and vertical parity. Horizontal parity array codes refer to array code methods in which the redundancy is independent of the data strips and is stored separately in redundant strips [21]. The parity data are stored separately in parity disks, and the original data are stored in the remaining disks. There is very good scalability in the entire disk array using this construction. Parity data in disks are used continuously in successive write operations, which always cause disk hotspots concentrated by continuous write operations. The tolerance of horizontal array codes is generally not very large. For example, the codes with two tolerances include EVENODD code [22], RDP code [23,24], Liberation code [25], and so on. Our team has conducted much related research already [26]. There is currently less research involving more than two tolerances, such as the STAR [27] code.

Some chunks store both original data and parity data in vertical parity array codes, which is different from that in horizontal parity array codes. The computational complexity overhead is distributed to each disk because of the simple geometric structure in vertical parity array codes. Unlike horizontal parity array codes, the bottlenecks caused by continuous write operations have been solved naturally due to its structural balance. However, the uniformity also leads to the strong dependence of disks on each other in vertical parity array codes, which leads to poor scalability [28].

### 2.4. Bit Matrix Coding

Bit matrix coding was first introduced in the paper on Cauchy Reed–Solomon coding as a parity check array coding technique [29]. CRS encoding is performed using the Cauchy distribution matrix to encode, and decoding is performed when the encoding fails [30]. Suppose in a finite field GF ($2^w$) [31,32], we expand the distribution matrix $(k + m) \times k$ with $w$ in both rows and columns to obtain a matrix with $w \times (k + m)$ rows and $w \times k$ columns, which we call the BDM (binary distribution matrix). The bottom $m \times (w \times w)$ matrix is called the CDM (coded distribution matrix). Matrix-based coding is essentially a dot-product operation of matrices and vectors, and different coding methods will result in different coding matrices [33]. The distribution matrix and the vector containing the data blocks are multiplied together to obtain a vector containing the data blocks and the check blocks. When encoding, the check block is obtained by the dot-product operation of the encoding matrix and the data vector. For decoding, we only need to find the rows in the distribution matrix corresponding to the data blocks that are not damaged and form a decoding matrix from these rows. By transposing this matrix and multiplying it with the vector composed of the data blocks that are not damaged, we can calculate the damaged data blocks. We expand the vector to obtain a vector containing $wk$ elements. Multiplying the matrix and the vector yields a vector containing $w(k + m)$ elements. This is shown in Figure 2. We can see that the bit distribution matrix is composed of many $w \times w$ submatrices.

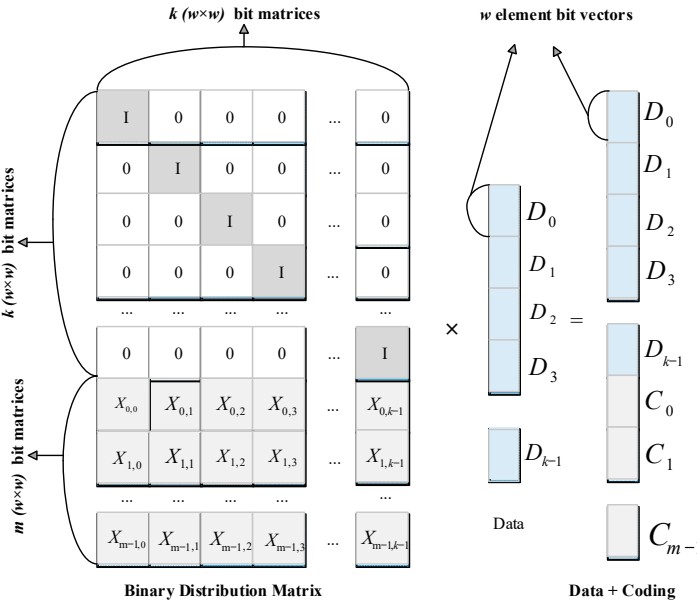

**Figure 2.** The encoding process is illustrated by the bit matrix–vector dot product.

In matrices and vectors, each element is a bit, and we can use the XOR operation instead of the dot-product operation [34]. This is accomplished by performing an XOR operation on the data vector corresponding to the bit that is one in the BDM. By using this approach, we can significantly increase the speed of encoding and decoding. Moreover, since the number of XOR operations performed is related to the number of ones in the BDM, the performance of encoding can be determined by the number of ones in the BDM.

## 3. NewLib Code

### 3.1. Encoding

#### 3.1.1. Encoding in the Liberation Code Pattern

This paper focuses on the design of big data coding schemes in cold data storage applications. Cold data refers to the data that users seldom access, which means that they always have a low utilization rate. If they exist in the form of replication, there is a waste of storage space. Moreover, the requirements are relatively low for the availability. In order

to minimize the use of the system resources, the speed of writing generally needs to be relatively high. Therefore, the encoding speed is the key consideration in the selection of coding schemes.

Suppose that there are data chunks (defined as *k*) and parity chunks (defined as *m*) in the coding system. We call this EC policy $k : m$. Each of the chunks has bits word length (defined as *w*). The erasure matrix can be described as $w \times (k + m) \times wk$ in the finite field $GF(2^w)$. Liberation, EVENODD, RDP, and Cauchy Reed–Solomon were selected to compare the performance of the erasure matrix. We selected the one with the best performance to use in our system.

Figure 3 shows the encoded parity matrix in Liberation code when k is 7 and *w* is 7. The black box represents one, while the white box stands for zero. It has been proved that the Liberation code is one of the MDS codes when $k < w$. The coding matrices for EVENODD, RDP, and Cauchy Reed–Solomon are shown in Figure 8a–c in the Plank article [25].

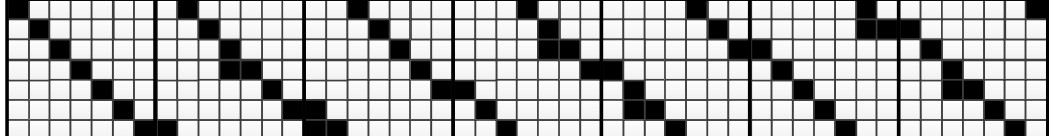

**Figure 3.** The erasure matrix in Liberation when k is 7 and w is 7.

For any given *w* and *k*, the number of ones in the parity matrix of the Liberation code is $kw + k - 1$. Since there are *kw* ones in the unit matrix of the BDM matrix, the total number of ones is $2kw + k - 1$ in the erasure matrix. The XOR operation number is determined by the number of ones. If the number of ones is *x* in an erasure matrix, the XOR operation number will be $x - 1$ in the encoding process. The number of XOR operations required in the Liberation code is shown as Equation (1):

$$\frac{2kw + k - 1 - 2w}{2w} = k - 1 + \frac{k - 1}{2w} \tag{1}$$

The optimal value of Equation (1) is $k - 1$.

We can obtain the number of XOR operations in the EVENODD code by the same method. This is shown as Equation (2):

$$\frac{kw + (w - 1)(k - 1) + kw - 2w}{2w} = \frac{3}{2}(k - 1) + \frac{k - 1}{2w} \tag{2}$$

As we can see, the number of XOR operations in the EVENODD code is almost one and a half times that of the Liberation code. In addition, we can use the ratio of ones in the parity matrix to compare the various codes. The proportion of ones of the Liberation code is 16%, as shown in Figure 3, while that of the RDP code is 28%, and that of the CRS code is 21%.

From the above comparison, we can see that the number of ones in the erasure matrix of the Liberation code is the lowest, which means that we can achieve the purpose of coding through fewer XOR operations in the actual coding process. Therefore, the coding scheme used in this paper is based on the Liberation code.

### 3.1.2. The Aligned Request

We take a request as an example to illustrate this situation. Suppose a request has an offset of 0 and a data size of 2 k.

From Figure 4, we can see that the EC policy is 4:2, and each strip data block size is 512 bytes. This task is divided into six requests after encoding, and each request is written to the corresponding node. If it is a write operation, all nodes will complete the task in cooperation. If it is a read operation, the data read from each node need to be transferred to the buffer of the user task in the reverse direction of the arrow. The data chunk and the parity chunk are not indicated in the figure, but there is an encoding process in the actual process.

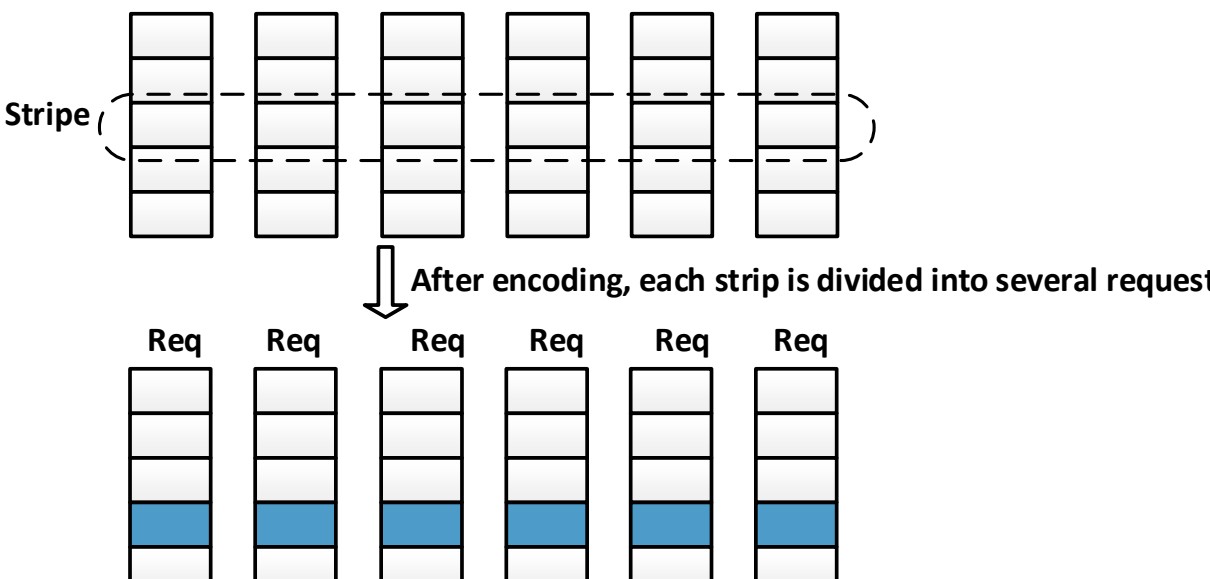

**Figure 4.** The aligned request.

### 3.1.3. The Non-Aligned Request

A non-aligned read request is different from an aligned read request. For non-aligned write requests, there may be an offset that is not aligned or the data size may not be aligned. The other data in the strip need to be read first. Then, they will be re-encoded and written back to the disk. The following is an example of offset non-aligned and data size non-aligned.

As shown in Figure 5, the non-aligned strip needs to be read and encoded together. A non-aligned write operation requires a partial read operation, which affects the performance of the system to some extent.

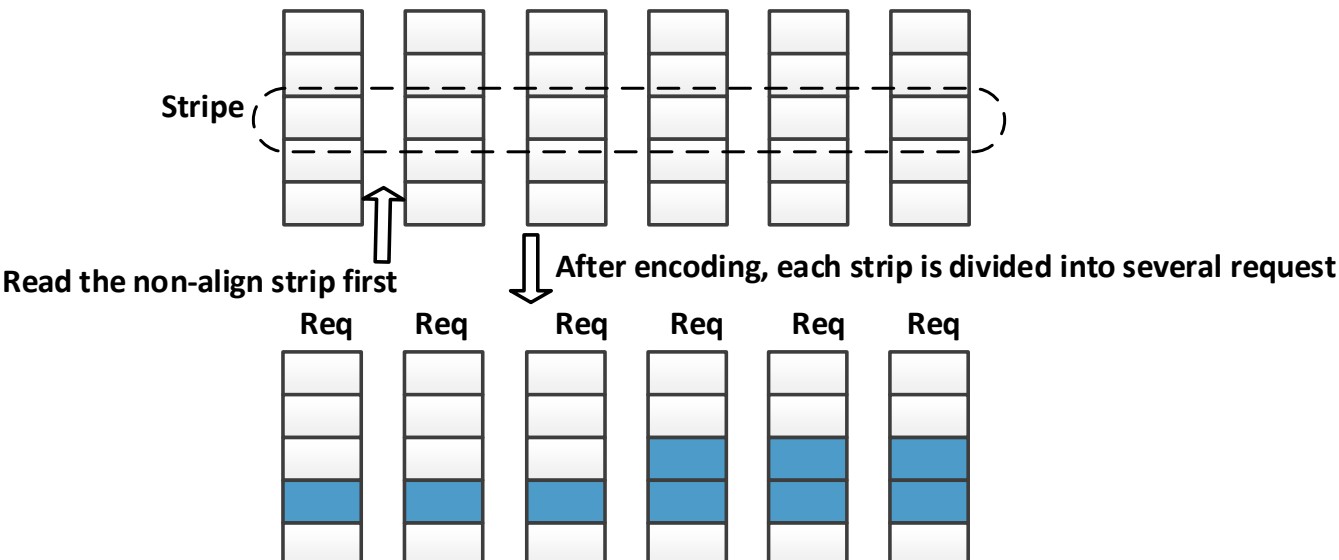

**Figure 5.** The non-aligned request.

### 3.2. Decoding in the N-Schedule Scheme

When a node leaves or data corruption occurs, a decoding operation is required to recover the lost data. In the following, we consider a case where an optimized decoding scheme is derived using an example.

Suppose $k = 7$ and $w = 7$. Suppose the information on data nodes $D_1$ and $D_2$ is lost; in order to recover the lost data, we take the first 14 rows of the check matrix and transpose them. With the obtained transposed matrix and the remaining data nodes, we can obtain the data on $D_1$ and $D_2$, which is shown in Figure 6.

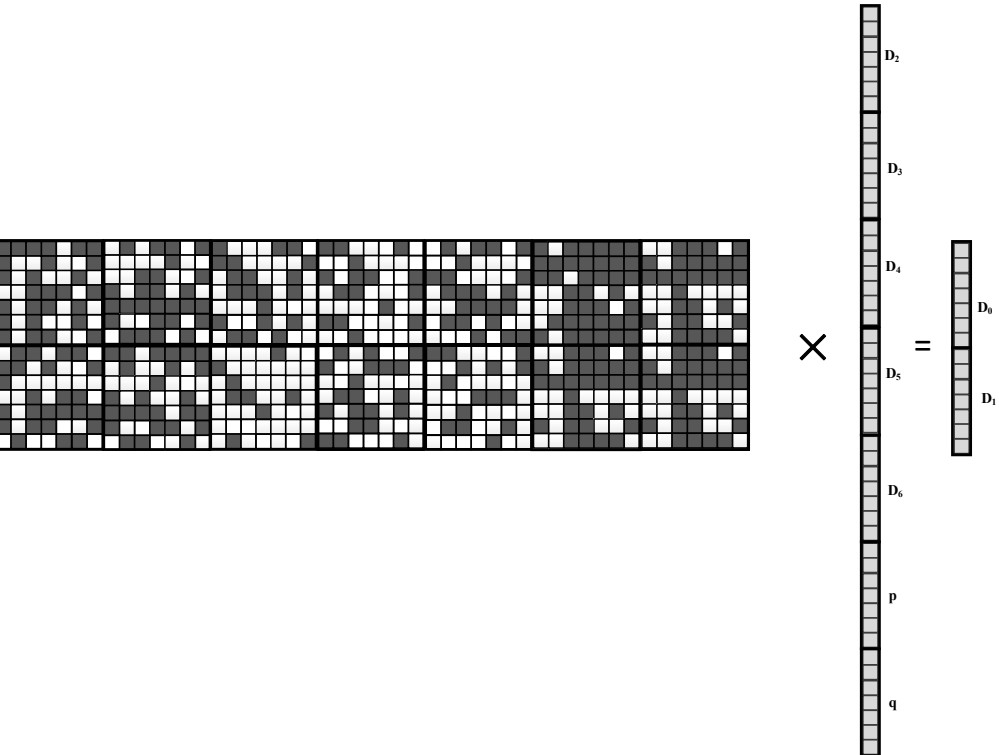

**Figure 6.** The decoding in the N-Schedule scheme.

The XOR operation number is equal to the number of ones minus one. The number of ones is 316 in 14 rows of the decoding matrix; hence, the number of XOR operations is 316 minus 14 which is equal to 302. Now, let us compare the number of ones in the first line with the number in the eighth line to calculate $D_{1,0}$ and $D_{2,0}$, respectively. The number of ones is 30 in the first line, while it is 25 in the eighth line, which means the number is the sum of 29 and 24. We can calculate one of the two lines from the XOR operation; then, the other line can be obtained from the first line using the erasure code method. We can formulate the equation shown as Equation (3).

$$d_{0,0} = d_{1,0} \oplus d_{2,0} \oplus d_{3,0} \oplus d_{4,0} \oplus d_{5,0} \oplus d_{6,0} \oplus p_0 \qquad (3)$$

There are only four XOR operations in Equation (3), which is the number of XOR operations to obtain $D_{1,0}$.

In summary, the other nodes are calculated by rationally utilizing the results that have been obtained, which can effectively reduce the XOR operation number. Let us continue with the example we just used, and with this optimized approach, the recovery process is as follows:

Calculate $d_{2,4}$: seven XORs.
Calculate $d_{1,4}$ from $d_{2,4}$: six XORs.
Calculate $d_{2,5}$ from $d_{1,4}$: seven XORs.
Calculate $d_{1,5}$ from $d_{2,5}$: six XORs.
Calculate $d_{2,6}$ from $d_{1,5}$: seven XORs.
Calculate $d_{1,6}$ from $d_{2,6}$: six XORs.
Calculate $d_{2,7}$ from $d_{1,6}$: seven XORs.
Calculate $d_{1,7}$ from $d_{2,7}$: six XORs.

Calculate $d_{2,1}$ from $d_{1,7}$: seven XORs.
Calculate $d_{1,1}$ from $d_{2,1}$: six XORs.
Calculate $d_{2,2}$ from $d_{1,1}$: six XORs.
Calculate $d_{1,2}$ from $d_{2,2}$: seven XORs.
Calculate $d_{2,3}$ from $d_{1,2}$: six XORs.
Calculate $d_{1,3}$ from $d_{2,3}$: seven XORs.

With this optimized decoding method, the XOR operation number required is 91, versus 302 in the original method. It can be seen that the efficiency of the decoding operation has been greatly improved.

## 4. Implementation

### 4.1. Architecture

We combined the erasure codes with replication for our system in this paper. Erasure codes are used to decrease the storage consumption, and replication can improve the read performance. The system framework is shown in Figure 7.

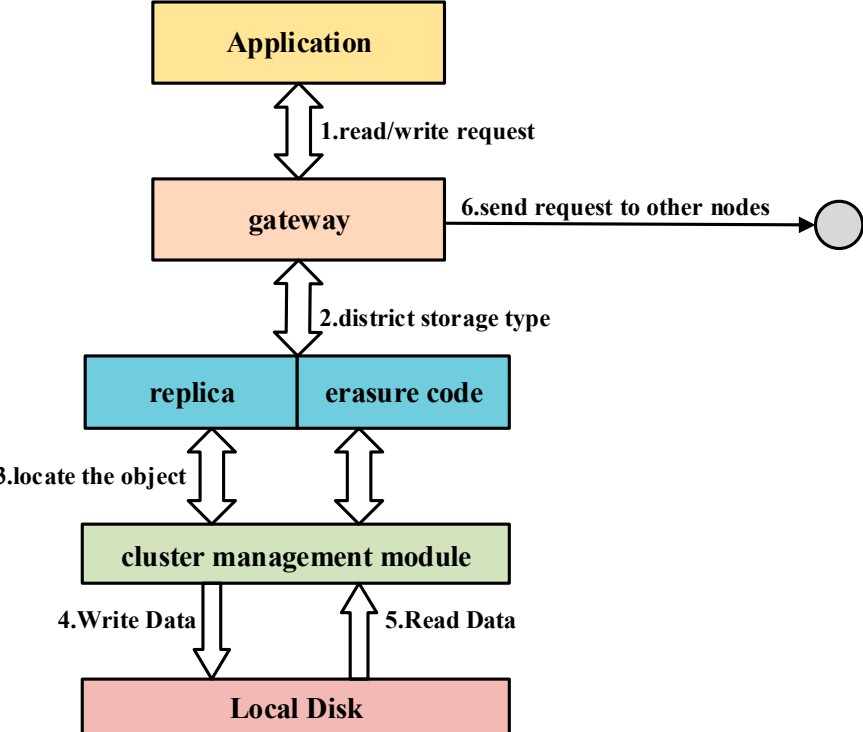

**Figure 7.** The architecture of the system.

After the read or write request is initiated by the application, it is sent to the gateway, which can judge whether the data block is located in the local server node or not. If it is located in the local server node, it will be classified into replication or erasure code according to the request type. Otherwise, it will be forward to other nodes.

The cluster management module is the most important module in the system. It can locate the object into the replication or erasure code. At the same time, it can maintain the consistency as soon as the node information has been changed in the system.

### 4.2. Node Schedule

Data addressing is one of the core technologies of file systems. Local file systems usually use metadata as an index to look up data. Metadata are usually stored on a logical disk or partition. A complete < block number, data > mapping table can usually be created with a simple data structure such as a tree or bitmap. Distributed file systems also require some mechanism for data addressing. If metadata are used for data addressing, the amount

of metadata in a distributed file system is so large that it cannot be fully stored on the local disk, requiring a separate node or cluster to store the large quantity of metadata.

A single node or cluster hosting metadata, while capable of achieving data addressing, has a serious drawback: the node where the metadata resides can become a bottleneck node. This is because all metadata requests are concentrated in a few nodes, which can lead to overloading of these nodes. If some of the metadata nodes lose data, it can seriously affect the performance of the cluster. To avoid this bottleneck, the underlying block storage system in this paper adopts a fully symmetric decentralized ring architecture with a consistent hashing algorithm to locate files in the storage nodes. It no longer has a super node or metadata server, and all nodes have equal status.

A ring topology is used for storage, and the nodes in the ring are divided into physical nodes and virtual nodes. The relationship between the physical nodes and virtual nodes is many-to-one, i.e., a physical node contains multiple virtual nodes, but a virtual node belongs to only one physical node. The node management module abstracts the physical nodes into multiple virtual nodes according to their storage and computing capabilities and obtains different IDs through consistent hashing algorithms based on the different IP address bits and port numbers of the physical nodes. These IDs are the unique numbers of different virtual nodes. The virtual nodes are evenly distributed to the storage ring according to the ID size. The virtualization of the physical nodes into multiple virtual nodes can play a role in load balancing.

So, how do we determine the relationship between the data blocks, nodes, and hash spaces? The following is an example, shown in Figure 8. Assuming that the range of the hash space is [0, 100), the ID values obtained by consistent hashing of the virtual nodes $A$, $B$, and $C$ are 0, 30, and 70, respectively, and the IDs along the clockwise direction until the next node is encountered belong to the hash space of this node; then, the hash spaces divided by the three nodes are $A[0, 30)$, $B[30, 70)$, and $C[70, 100)$, respectively. By calculating the consistent hash value of a data block, we can determined in which virtual node the data block should be stored. When a node, say node $B$, hangs, the hash space is redistributed at that point, and the hash spaces of nodes $A$ and $C$ are then $A[0, 70)$ and $C[70, 100)$. The hash space of node $B$ is redistributed to $A$, and at the same time the data on $B$ are transferred to node $A$. However, if virtual node $B$ and virtual node $A$ are located on the same physical node, then the hash space of $B$ is not allocated to $A$ but to $C$ (assuming that $B$ and $C$ are located on different physical nodes).

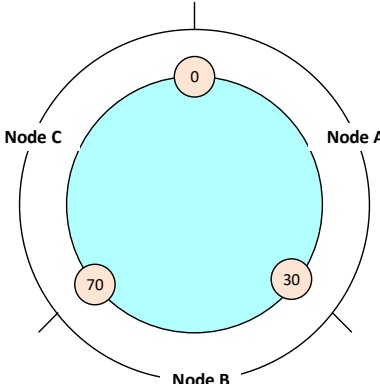

**Figure 8.** The correspondence of the data blocks, nodes, and hash spaces.

## 5. Evaluation

### 5.1. Experimental Setup

We used FIO in the tests, which is a very useful tool to test the IOPS. It is mainly used for pressure testing and verification of the system. It can simulate different IO operations through multiprocessor or multithread, which can support nineteen different I/O engines,

including Sync, MMAP, SG, and so on. It can run not only on the block storage device but also in the file storage device.

In the test, we tested sequential reading, random reading, sequential writing, and random writing using the two indicators IOPS and throughput. Since the size of blocks is the important variable to influence the performance, we set the block size to be 4 K, 64 K, 512 K, and 1028 K, respectively.

### 5.2. Read Performance Test

#### 5.2.1. The Performance in Sequential Reading

The result of the IOPS in sequential reading is shown in Figure 9. The value of the IOPS decreased as the size of the data block increased, since the IOPS of the system was inversely proportional to the time reading each block, which increased as the block size increased. When the block was 4 K, the value in the NewLib code was about 25% lower than that in the triple replica.

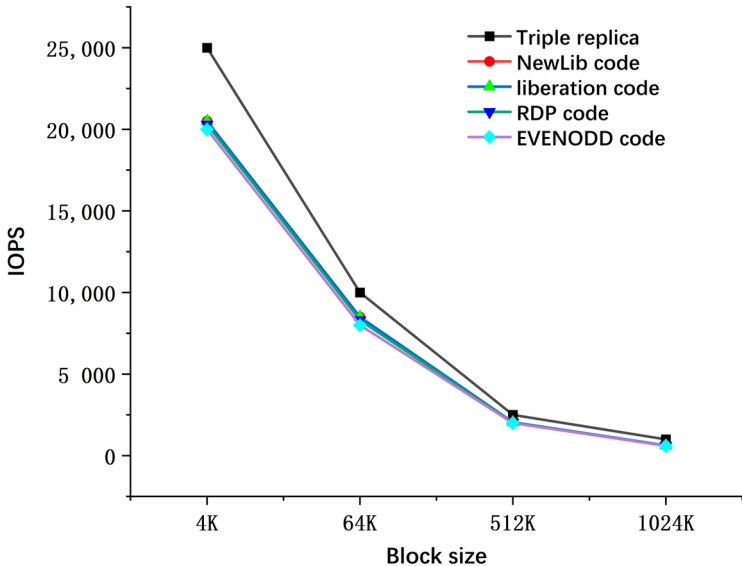

**Figure 9.** The IOPS in the sequential reading.

The throughput value in the sequential reading changed with the block size and backup mode, which is shown in Figure 10. We can see that as the data block increased, the value of the throughput also increased. This is because after the data block became larger, each read could read more data while minimizing the metadata overhead and disk movement overhead. Reading the same quantity of data, the smaller the data block, the greater the metadata processing overhead required. For a fixed block size, we can see that the copy mode still had some advantages over the EC mode when reading data.

From Figures 9 and 10, we can see that as the data block increased, the IOPS decreased steadily, but after the throughput increased to a certain extent, the value remained basically stable. For the data blocks of 512 K and 1024 K, whether in copy mode or in erasure code mode, the throughput values were basically equal. This is because once the data block was large enough, its size had little impact on the overhead of the metadata.

#### 5.2.2. The Performance in Random Reading

The above tests were for the sequential reading performance. Next, we examined the performance of the system under different patterns and block sizes for random reads.

From Figure 11, we can see that the trend for random reading was consistent with the trend for sequential reading. The larger the data block was, the lower the value of the IOPS. When we fixed the data block size to compare the different backup strategies, we can see that the replica strategy had certain advantages in terms of the IOPS relative to the erasure coding strategy. The test results were also consistent.

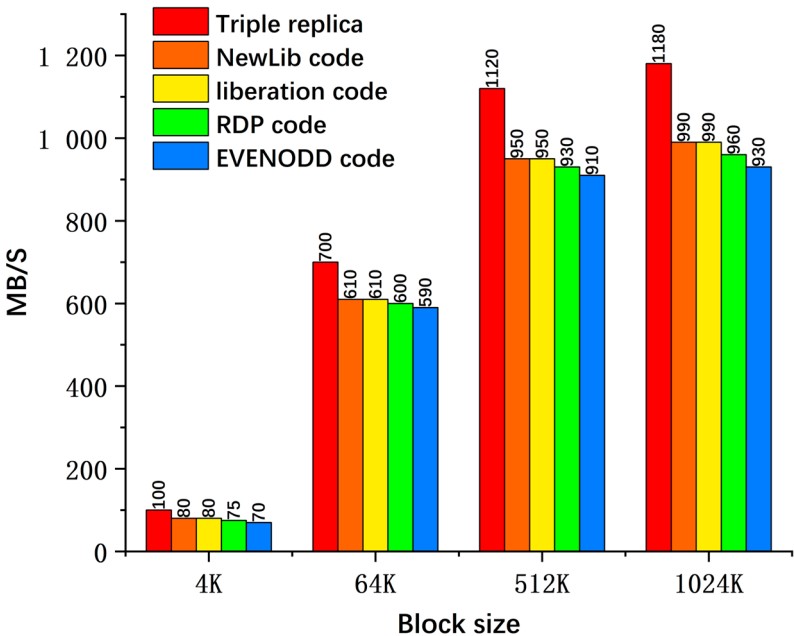

**Figure 10.** The throughput in the sequential reading.

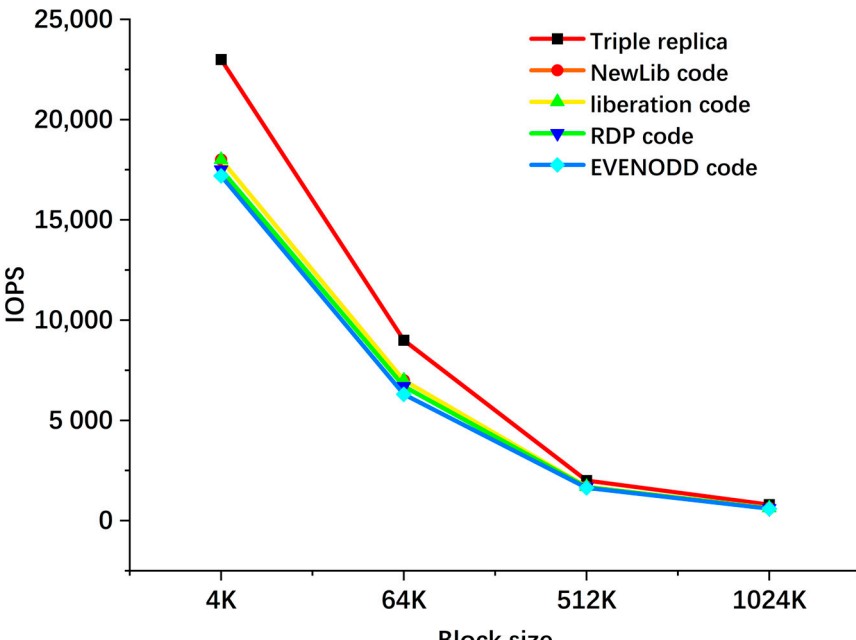

**Figure 11.** The IOPS in the random reading.

Figure 12 shows the throughput value for a random read. In the figure, we can see that the throughput value increased as the data block increased, but it was consistent with the sequential read. After the data block size reached 512 K, the change in the data block size had little influence on the value of the throughput. The throughput values were basically equal with data blocks 512 K and 1028 K, whether they were replica mode or erasure code mode.

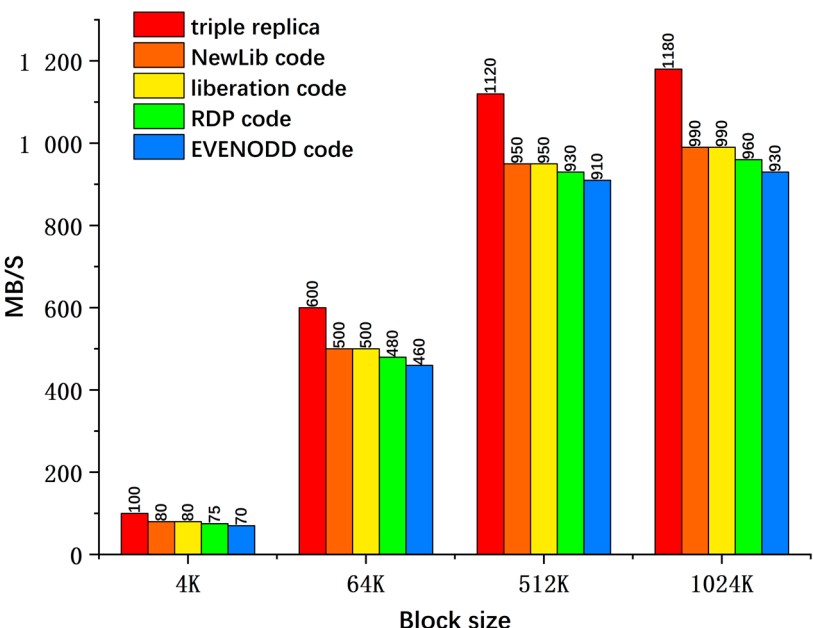

**Figure 12.** The throughput in the random reading.

Through the above four tests—the sequential read IOPS and throughput and the random read IOPS and throughput—we can see that the read speed of the replica mode was better than that of the erasure code mode, which was consistent with the theoretical value. This is because the replica mode can select the data block of the machine for reading, while the machine only stores part of the data and needs to call the data stored in other nodes through the network in the erasure code mode. At the same time, we can also see that the size of the data block had a great influence on the IOPS and the throughput value. Therefore, in conducting research to improve the performance of the storage system, it is important for us to set a reasonable data block size to improve the read performance of the system.

### 5.3. Write Performance Test

The focus of this paper is the research of erasure code technology for cold data in big data systems; therefore, the data writing performance is a key factor. Cold data are usually large quantities of data written sequentially at one time and rarely change; so, we focused on the sequential write performance of the massive data erasure code in the cold backup strategy. The random write function will be implemented in future work. The writing performance tests were also divided into two categories: the IOPS and the throughput.

The result of the IOPS in sequential writing is shown in Figure 13. From the figure we can see that the trend for the sequential writing was quite different from that for the sequential reading.

Compared to the IOPS of the sequential reading operation, the IOPS of the sequential writing operation was significantly smaller than that in the replica mode. This is because the writing operation required more metadata and disk operations relative to the read operation. More importantly, in the case of a triple replica, the writing operation generated two times more network bandwidth than the quantity of data written.

Figure 14 shows the throughput of a sequential write operation. We can see that the quantity of data written became larger as the number of data blocks increased. However, once the data block was larger than 64 K, the growth tended to be flat. Unlike the read operation, the throughput of the erasure code write operation was larger than the throughput of the copy write operation.

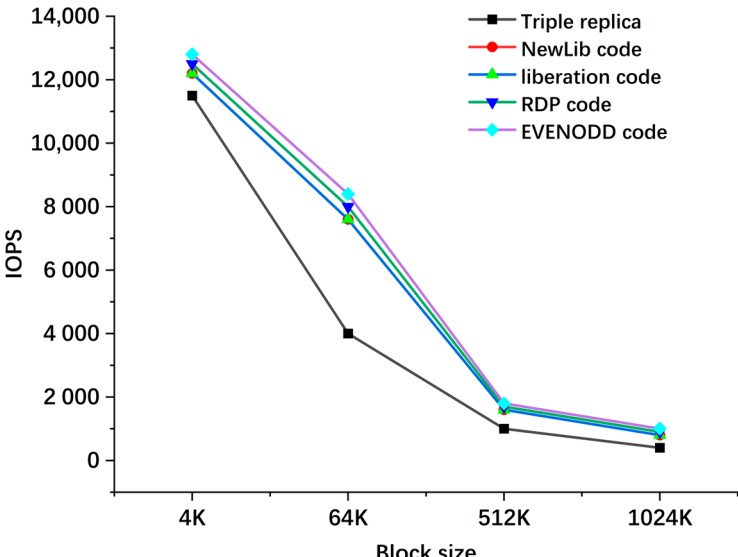

**Figure 13.** The IOPS in the sequential writing.

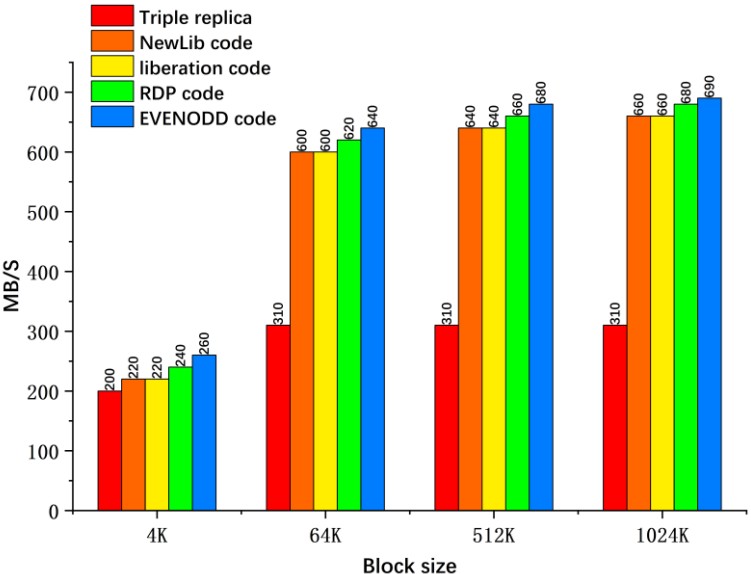

**Figure 14.** The throughput in the sequential writing.

## 6. Conclusions

In order to improve the encoding and decoding speed of distributed storage systems, especially for "cold" data, we proposed a new erasure code process, the NewLib code, which can strip the encoding data to data alignment. The NewLib code improved the performance of reading and writing in the distributed storage systems. For uniform data distribution and task scheduling, we developed the N-Schedule strategy. The data nodes were divided into multiple virtual nodes, and the virtual nodes were dispersed to a hash ring through consistent hashing to construct a fully symmetric and decentered hash ring.

In the future, we need to address the correct application of the generalized delete code in big data storage systems to improve the decoding speed. By increasing the decoding speed or adjusting the decoding time, the corrective censoring codes can be applied to "hotter" data. In this way, the strategy of copying data can be completely discarded so as to improve the efficiency of the storage system.

**Author Contributions:** Conceptualization, C.Y.; Data curation, S.Y.; Formal analysis, W.L. and T.L.; Funding acquisition, C.Y.; Investigation, S.Y.; Methodology, C.Y.; Resources, C.Y. and Y.L.; Software, W.L. and T.L.; Supervision, Y.L.; Validation, C.Y. and Z.X.; Visualization, S.Y.; Writing—Original draft, Z.X.; Writing—Review and editing, C.Y. and Z.X. All authors have read and agreed to the published version of the manuscript.

**Funding:** This research was funded by the National Natural Science Foundation of China, grant number 61662038, and in part by the fund of the Science and Technology Project of Jiangxi Provincial Department of Education, grant number GJJ211816.

**Institutional Review Board Statement:** Not applicable.

**Informed Consent Statement:** Not applicable.

**Data Availability Statement:** Not applicable.

**Conflicts of Interest:** The authors declare no conflict of interest.

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
