# Peer review of "Erasure Codes for Cold Data in Distributed Storage Systems"

_applsci, doi:10.3390/app13042170_

Round 1
Reviewer 1 Report
The topic of this paper is difficult to extract from the title and the abstract. The entire text looks like a collection of randomly selected phrases from the area of distributed data storage. As one of two main contributions, the authors claim a new coding scheme but indeed they simply list known approaches and select among them one well-known solution.
The paper is extremely poorly written, it contains many typos and non-understandable statements. Some phrases are very unexpectable, for example, see p.3, lines 4-5:
“Service disruptions caused by sexual failure, and many data centers are inadequate for failures [20]. “
Reviewer 2 Report
I suggest that the authors replace some images that are not of good resolution with high resolution images.
Reviewer 3 Report
The main objective of this paper is to improve the availability and reliability of such systems, as well as to maximize the utilization of storage space. The scheme is based on a combination of two strategies: erasure codes technology and data copying strategy (replica). Test results show that this scheme can improve system performance.
Figure 1. Google cloud storage system architecture - needs to be improved it's totally unclear.
Figure 9. The architecture of the system. - Needs to be improved.
4.2. Node schedule - needs to be explained into more details.
Minor spellchk errors schould be corrected.
Conclusion would benefit from rewriting.
Proposed scheme NewLib code and P-schedule strategy developed with the aim of improving the speed of encoding and decoding in distributed storage systems, with a special focus on the so-called "cold" data. This scheme deals with solving the problem of data redundancy and its availability. In the future, the authors state that the application of generalized delete code technology in large data storage systems should be considered, which would further improve the decoding speed. Thus, it could also be applied to "hotter" data, which would make it possible to completely abandon the strategy of copying data and improve the efficiency of the storage system.
Reviewer 4 Report
References from [25] are skipped.
Section 3 should be rewritten, making sure to cite Planck's paper. Authors should evaluate the gain of the generalized decoding method with respect to its parameters and should provide another example if they wish.
Round 2
Reviewer 1 Report
1. I did not receive the corrected version. The current three-color version is unreadable.
2. In error-correcting coding theory the generator and parity-check matrices are presented in formulas, not pictures.
3. Coding theory literature on erasure-correction codes is almost wholly ignored. Only Reed-Solomon codes are mentioned.
Reviewer 4 Report
The authors do not properly cite Plank's paper [25]. For instance, their Figure 3 is exactly Figure 9 in [25]. They also do not point out that algorithm for decoding used by them in Section 3.2 is in fact the algorithm described in Section 3.4 of [25].